# Service access experiences of immigrant and refugee caregivers of autistic children in Canada: A scoping review

**Lina Marie Casale**[1]*, **Stephen J. Gentles**[2], **Janet McLaughlin**[3], **Margaret Schneider**[4]

**1** Department of Health Studies, Faculty of Human and Social Sciences, Laurier Autism Research Consortium, Wilfrid Laurier University, Brantford, ON, Canada, **2** Department of Health Studies, Faculty of Human and Social Sciences, Wilfrid Laurier University, Brantford, ON, Canada, **3** Department of Health Studies, Faculty of Human and Social Sciences, Co-Director, Laurier Autism Research Consortium, Wilfrid Laurier University, Brantford, Ontario, Canada, **4** Department of Kinesiology and Physical Education, Faculty of Science, Co-Director, Laurier Autism Research Consortium, Wilfrid Laurier University, Waterloo, Ontario, Canada

* casa4310@mylaurier.ca

## Abstract

Primary caregivers are the main mediators of care for children with an autism diagnosis in Canada, and the navigation process to gain access to autism-related services is known to be a major burden. These challenges to service access are compounded for newcomers to Canada, which include immigrants and refugees. The purpose of this scoping review is to describe the available research on Canadian newcomer caregiver experiences navigating and accessing autism-related services. After a systematic search and screening process, 28 studies were included. Data were extracted regarding the populations, study aims, and themes reported. Included studies characterized barriers and facilitators to service access and navigation specific to immigrants, while limited information was available for refugees. Based on the existing literature, the authors provide recommendations for possible research approaches, populations to include, and themes to examine in future research to promote health equity in Canadian autism service access.

## 1. Introduction

Autism (autism spectrum disorder, or ASD) is a heterogenous, lifelong, neurodevelopmental condition whose prevalence is not considered to be affected by race, ethnicity, culture, or socioeconomic status [1]. In 2018, the Autism and Developmental Disabilities (ADDM) Network estimated that 1 in 44 eight-year-old children were diagnosed with autism [2]. Parents or other primary caregivers (hereafter caregivers) commonly pursue numerous interventions for a broad range of possible autism-related and comorbid concerns over the course of their child's development [3, 4]. However, given the variability in how autism presents, differing interventions and supports, alongside inconsistent availability and funding for different services across provinces, territories and regions in Canada, caregivers have no uniform, prescribed approach to follow. The onus is therefore on caregivers to research potential

**Data Availability Statement:** All relevant data are within the paper and its Supporting Information files

**Funding:** JM received an Individual Partnership Engage Grant through the Social Sciences and Humanities Research Council (SSHRC) (grant no. 892-2020-3083) (website: https://www.sshrc-crsh.gc.ca/funding-financement/programs-programmes/partnership_engage_grants-subventions_d_engagement_partenarial-eng.aspx) and a SSHRC Individual Partnership Development Grant (grant no. 890-2021-0088) (website: https://www.sshrc-crsh.gc.ca/funding-financement/programs-programmes/partnership_development_grants-subventions_partenariat_developpement-eng.aspx). LMC also received financial support from Wilfrid Laurier University. The funders had no role in the study design, data collection and analysis, decision to publish, or preparation of the manuscript.

**Competing interests:** The authors have declared that no competing interests exist.

interventions and service providers themselves, which includes learning how to access them [4, 5]. The burdens of navigating services common to most English-speaking countries include time needed to research care, the substantial financial cost for private options, travel to frequent appointments, coping with barriers such as waitlists, and the complexity of seeking services across multiple care systems (e.g., health, education, and social care) [3, 6, 7].

These burdens can impact caregivers' mental and physical health in significant ways, including physical and psychological exhaustion, family dysfunction, and deteriorating physical health [3]. For example, the financial burdens attributable to pursuing private therapy, travel to services, and lost employment income due to caregiving demands, have important impacts on stress [8]. In a qualitative study, the overall stress of service navigation was reported by caregivers to be a major source of mental health problems, such as anxiety and depression [3].

While service accessibility and navigation challenges are common for caregivers of all backgrounds, newcomers (i.e., immigrants and refugees) face compounded difficulties, given their unique experiences. For instance, a family's culture can impact their view of disability, which may be different from the dominant norms within the host society [9–11]. Further, caregivers may be unaware of or uneducated about autism, and their cultural understandings of autism may influence service-seeking and treatment decisions [10]. Reviews conducted outside of Canada indicate that newcomers may face language barriers [9, 11], fragmented services [9], difficult administrative processes [10], or a lack of providers [9]. These reviews also describe how evidence exists of delayed autism diagnosis among ethnic minorities [10], as well as that culturally and linguistically diverse families also face a greater risk of not obtaining early intervention services [9]. Newcomers overall may also lack familial support after immigrating [12]. Immigrants may also face economic barriers stemming from the cost of immigration or low-paying jobs upon arrival [11], which could affect service affordability. Jennings and colleagues also note that medically disabled refugee children in Canada are in a particularly vulnerable position and, in addition to facing similar barriers to other immigrants, these families may have endured complex trauma [12]. Canada is a good place to examine issues facing newcomers, given that as of 2016, 37.5% of Canadian children are either first or second-generation immigrants, and this proportion is anticipated to grow [13]. Additionally, as of 2016, Canada was home to over 95,000 refugees under the United Nations High Commissioner for Refugees' (UNHCR) mandate [14]. This population's underrepresentation indicates a gap in Canadian autism research.

There are aspects of the newcomer experience that overlap with that of other groups in Canada, including the effects of racialization and ethnocultural difference. In a scoping review examining minority health, Khan and colleagues note a gap in Canadian health research on visible minorities in Canada, with existing studies often not making the distinction between visible minority (or racialized) populations born in Canada and visible minority *immigrants* [15]. This can obscure inequalities among Canadian-born visible minorities due to the healthy immigrant effect (which refers to the phenomenon of recent immigrants being healthier than their host country's general population) [15, 16], and emphasizes the need to examine studies with distinctive findings between these groups. Moreover, newcomers have often been underrepresented across multiple areas of Canadian autism research [10, 11, 17–21]. This includes a lack of research on culturally specific knowledge of autism, as well as how culture impacts preferences for different supports [22]. To date, precise data on autism prevalence among immigrants and refugees in Canada is also limited, although the 2019 Canadian health survey on children and youth did not find any statistically significant difference in autism prevalence by population groups, including visible minority status [23]. That said, there is evidence from other countries which suggests that immigrant groups may experience higher rates of autism prevalence; for example, Bolton and colleagues observed higher rates of autism among a

migrant population in Ireland composed primarily of children from Sub-Saharan Africa [24], and an American study conducted by Becerra and colleagues noted that the race, ethnicity, and nativity of mothers was associated with various aspects of their child's autism diagnosis [25]. These disparities, as well as data from other countries suggesting that autism presentation itself among these groups warrants further study, suggest the need to examine the unique service and support needs of newcomers to Canada.

The purpose of this scoping review was to describe the available research on the experiences of Canadian newcomer caregivers of autistic children with respect to navigating and accessing autism-related services, and to identify research gaps. Existing reviews on the experiences of immigrant or refugee caregivers are based in the United States (e.g., [9, 11]), are not focussed solely on Canadian research (e.g., [10]), or focus broadly on caregivers of all disabled children (e.g., [12]). Our intention with this review was to provide a summary of the literature that can be used in emerging policy work promoting health equity in autism services in Canada (e.g., [26]) and other countries with growing newcomer populations, and to inform future research including methodology.

This review (search strategy, extraction, analysis) was guided by the question: What do Canadian newcomer caregivers of autistic children experience (including barriers) when navigating and accessing autism services (including diagnostic services)?

## 2. Methods

We followed scoping review methods described by Arksey and O'Malley [27] and Peters and colleagues [28]. We also followed the Preferred Reporting Items for Systematic Reviews and Meta-Analyses (PRISMA) standards for reporting scoping reviews [29]. The search was developed according to the breakdown of the populations represented in the research question (i.e., terms relevant to autism, caregiving, and immigrant/refugee status were included)- see S1 Appendix. The search was restricted to include English-language studies published from the year 2000 onwards, since only literature providing a current understanding of the phenomenon of interest was considered relevant given the rapid evolution of autism-related diagnosis and service contexts. Searches were conducted by two independent reviewers in August of 2023. The search was implemented in the following databases: CINAHL on August 21st, 2023 (EBSCO), Sociological Abstracts on August 21st, 2023 (ProQuest) PsycINFO on August 10th, 2023 (ProQuest), and MEDLINE on August 21st, 2023 (ProQuest). See S1 Appendix for the search strands used in each database. The bibliographies of the studies included from the database searches were also examined for relevant articles [27]; this was continued until reference lists no longer yielded novel, relevant titles.

Eligibility was determined in two steps by two independent reviewers: first, titles and abstracts were screened, followed by screening of the retrieved full text. Articles were retained following the screening of their titles and abstracts if the study focused on immigrant or refugee caregivers of autistic or developmentally or intellectually disabled individuals, or professionals who worked with these populations, as well as if the article focused on service access, navigation, and/or broad experiences of newcomer caregivers of autistic children in Canada. Articles were excluded if the title or abstract indicated the study took place outside of Canada, the population focus was on minority groups broadly as opposed to newcomers, or if the study focussed on autism prevalence, language-learners, the usage or effectiveness of medications or therapies, or suspected causes of autism.

At the eligibility stage wherein full-text was assessed, the following eligibility criteria were applied:

Inclusion criteria:

1. The study population included at least one autistic individual, one caregiver of an autistic individual, or professionals who work with these individuals

2. Any caregivers of autistic individuals included in the study were immigrants or refugees

3. Any autistic individual included in the study population was a refugee or a first- or second-generation immigrant focused on individuals with or caregivers of individuals with autism, developmental, and/or intellectual disabilities

4. The article focused on service access, navigation, and/or broad experiences of the aforementioned population

Exclusion criteria:

1. The study was not conducted in Canada

2. The study population included minority groups but not specifically immigrants or refugees

3. The study population did not include at least one participant who worked with or was a caregiver of an autistic individual

4. The study was on autism prevalence, language-learners, the usage or effectiveness of medications or therapies, or suspected causes of autism

Data were extracted by one reviewer (LMC) on the study characteristics, study aims, identified barriers and facilitators to service access, qualitative themes, and conclusions. Study characteristics included participants' countries of origin, participants' caregiving roles, methodological approach, study location within Canada, and the language(s) used in data collection. Categories for grouping the studies based on similarities in their aims were developed as they were reviewed. Extracted data were entered and maintained in an Excel database.

Subcategories within the broader themes of service access barriers and facilitators were developed to reflect the shared, specific experiences of participants across studies during the iterative process extraction and analysis. For instance, within the theme of "Barriers to service access," one of the subcategories that emerged was "Barriers related to coming to a new country." The titles and scope of these subcategories were iteratively refined as common and novel themes were mapped out across studies and then sorted into broader categories. The process of identifying key themes from studies was completed by the first author with assistance from a non-author assistant to prevent bias in the creation of categories and to help ensure that themes were arising independently within the literature.

## 3. Results

After removal of duplicates, the database search yielded 237 records (See Fig 1). This, combined with a total of 1241 novel results from three rounds of bibliography searches, meant the *Identification stage* yielded 1478 results for screening. The high quantity of results identified from other sources (bibliographies of included studies), as opposed to through database searches, can be attributed to the observation that many articles did not specify study location despite appearing to meet all other inclusion criteria; many studies were found to have been conducted in countries outside of Canada upon further review. After title and abstract screening, 132 articles were retained. Upon screening full text, 102 articles were excluded (reasons for exclusion included the study not being novel, not being Canadian, not being specific to the target populations, or not being written in English- see Fig 1). Ultimately, 20 studies from database searches were included and 8 studies from bibliography searches were included in this review, resulting in 28 studies total.

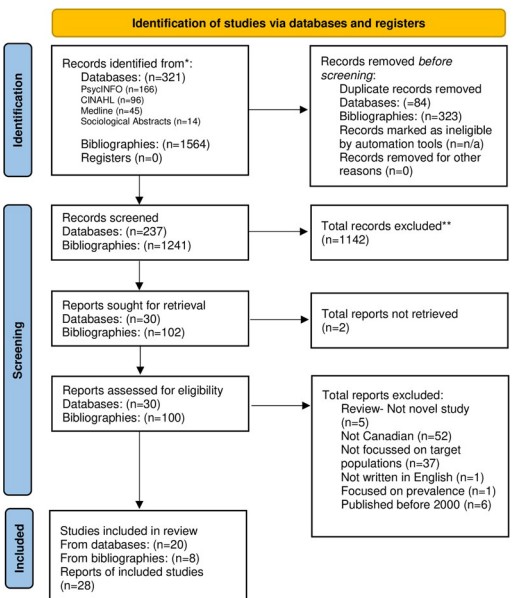

**Fig 1. Completed PRISMA flowchart (format based on Peters and colleagues [24], chart adapted from Page and colleagues [29]).** *Consider, if feasible to do so, reporting the number of records identified from each database or register searched (rather than the total number across all databases/registers). **If automation tools were used, indicate how many records were excluded by a human and how many were excluded by automation tools.

### 3.1. Study characteristics

Newcomer caregivers of autistic children were included as participants in all 28 of the included studies [17–21, 30–52], and two of the 28 studies on caregivers specifically cited the experiences of refugee caregivers [31, 49]. Fig 2 includes, for reference, a comparison of the percentage of the total number of countries of origin identified in the studies to the population-wide data on the place of birth of all immigrants who arrived in Canada from 2001–2021 [53]. Implications of this comparison are discussed in subsequent sections. Twelve studies focused on specific cultural and ethnic groups [18, 20, 31–35, 37, 38, 44, 48, 51], while 13 studies examined the experiences of newcomers from a variety of ethnic and cultural groups [19, 21, 30, 41–50]. Additionally, eight studies included both Canadian-born and immigrant caregivers of autistic children as participants [17, 34, 36, 38–41, 52].

While not every study indicated whether the responding caregivers identified as mothers or fathers, when the distinction was made most caregivers were mothers. Eight studies focused explicitly on mothers [18–20, 37, 38, 48, 49, 51], whereas none focused explicitly on fathers. The differences in the perspectives between mothers and fathers were explicitly distinguished in five studies [21, 40–42, 45]. No studies identified extended family members as primary caregivers. Two studies included the perspectives of professionals working with newcomer caregivers [31, 50].

Fifteen studies used qualitative methodologies [18–21, 31, 32, 34, 37, 42, 43, 45, 48–51], five used quantitative methodologies [36, 39–41, 52], and eight used mixed methods [17, 30, 33, 35, 38, 44, 46, 47]. In-depth interviewing was the qualitative method used in 13 studies [17–21, 31, 35, 42–45, 49, 51]. Other methods used included field notes, focus groups, and critical personal narratives. Data were gathered in English in 18 studies [19, 21, 31–35, 39, 41–50], French in 10 studies [21, 30, 40–43, 45–47, 49], Spanish in five studies [21, 42, 43, 47, 49], Chinese dialect in three studies [20, 50, 51], Portuguese in two studies [42, 43], Korean in two studies [32, 33],

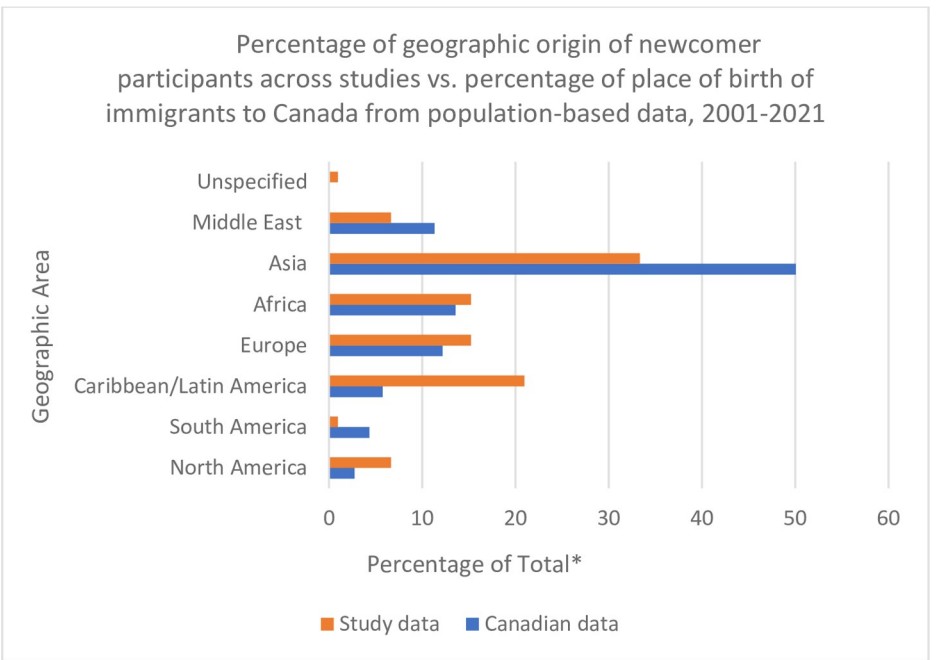

**Fig 2. Comparison of country of origin of newcomers as a percentage of the total number of countries of origin identified in the studies to population-based data on place of birth of Canadian immigrants from 2001–2021 (included for reference).** *Note: For study data, the percentage on the X-axis represents a percentage of the total number of countries of origin identified across all studies included in the review. For Canadian data, it is a percentage of the total number of immigrant to Canada between 2001–2021 as reported by Statistics Canada. [Camard et al., 2022; Decoteau, 2017; Fong et al., 2021; Fong et al., 2022; Fong et al., 2023; Fontil & Petrakos, 2015; Grewal, 2010; Kediye et al., 2009; Khanlou et al., 2017; Kuan et al., 2022; Lai & Ishiyama, 2004; Lee & Zhu, 2021; Luthra, 2010; Luthra, 2018; Millau et al., 2016; Millau et al., 2018; Millau et al., 2019; Pondé & Rousseau, 2013; Pondé et al., 2019; Ravindran & Myers, 2012; Rivard et al., 2019; Rivard et al., 2020; Rivard et al., 2021; Shanmugarajah et al., 2022; Skrinda, 2008; Starr et al., 2016; Statistics Canada, 2023; Su et al., 2021; Weiss et al., 2016].

and Arabic in one study [50]. Several studies used interpreters for interviews conducted in other languages [31, 42, 43]; Luthra noted using interpretation services to assist participants with questionnaire completion [38]. The specific language being used by participants was not specified in five studies [17, 18, 36, 37, 52].

Provincial study locations within Canada included 11 from Ontario [18, 19, 31, 36, 38, 39, 48–52], nine from Quebec [17, 21, 30, 40–42, 45–47], six from British Columbia [20, 32–35, 37], one from Alberta [35], and one did not specify [44]. Some studies specified regions within provinces, including eight within the Greater Toronto Area [18, 19, 31, 36, 38, 48, 49, 51], and seven in the Montreal area [21, 41–43, 45–47].

### 3.2. Study aims

Study aims were extracted based on the articles' research questions, objectives, and target populations. They were then coded and sorted into qualitatively derived categories within Excel. A total of 11 categories were developed from this initial data review, ten of which were present across multiple studies. The most prevalent aim noted in 11 studies was to learn more about the general barriers and stressors experienced by newcomer caregivers which, for the purposes of this review, includes evaluations of overall quality of life [18, 19, 34, 36, 38, 40, 41, 48, 49, 51, 52]. A full breakdown of the included studies' aims can be found in Table 1.

**Table 1. Categorization of study aims.**

| Categories of study aims | Number of studies | Percentage of studies | Studies |
|---|---|---|---|
| Perception of/feelings toward autism | 9 | 31% | Decoteau, 2017; Luthra, 2010; Millau et al., 2018; Pondé & Rousseau, 2013; Pondé et al., 2019; Ravindran & Myers, 2012; Rivard et al., 2019; Starr et al., 2016; Su et al., 2021 |
| Understand caregivers' treatment priorities | 1 | 3% | Millau et al., 2018 |
| Gain insight into professional perspectives | 2 | 6% | Decoteau, 2017; Starr et al., 2016 |
| Examine experiences pertaining to racism or "othering" | 2 | 6% | Decoteau, 2017; Skrinda, 2008 |
| Gain insight into existence of service barriers | 9 | 31% | Camard et al., 2022; Fong et al., 2022; Fontil & Petrakos, 2015; Grewal, 2010; Khanlou et al., 2017; Rivard et al., 2019; Rivard et al., 2020; Rivard et al., 2021; Skrinda, 2008 |
| Gain insight into existence of service navigation facilitators/support services | 7 | 24% | Camard et al., 2022; Fong et al., 2022; Fontil & Petrakos, 2015; Rivard et al., 2019; Rivard et al., 2020; Rivard et al., 2021; Su et al., 2021 |
| Examine the use of/feelings towards specific interventions | 3 | 10% | Ravindran & Myers, 2012; Rivard et al., 2021; Shanmugarajah et al., 2022 |
| Understand caregivers' experiences in the education system | 3 | 10% | Fontil & Petrakos, 2015; Lai & Ishiyama, 2004; Starr et al., 2016 |
| Understand caregivers' experience of the diagnostic process | 6 | 21% | Grewal, 2010; Lee & Zhu, 2021; Millau et al., 2018; Pondé & Rousseau, 2013; Rivard et al., 2019; Shanmugarajah et al., 2022 |
| Inquiry into existence of general stressors/challenges | 11 | 38% | Fong et al., 2021; Kediye et al., 2009; Khanlou et al., 2017; Kuan et al., 2022; Luthra, 2018; Millau et al., 2016; Millau et al., 2019; Shanmugarajah et al., 2022; Skrinda, 2008; Su et al., 2021; Weiss et al., 2016 |
| Inquiry into existence of buffers for general stressors | 7 | 24% | Fong et al., 2021; Fong et al., 2023; Kuan et al., 2022; Luthra, 2018; Millau et al., 2019; Shanmugarajah et al., 2022; Weiss et al., 2016 |

### 3.3. Themes

**3.3.a. Barriers to accessing adequate services.** Fourteen of the included studies identified barriers to accessing adequate services [17, 19, 30, 32, 33, 35, 38, 41, 45–49, 51]. This included difficulty accessing transportation to services [19, 33, 35, 47, 49, 51], long waitlists [19, 30, 32, 38, 45, 49, 51], the disbursement and fragmentation of services [19, 51], and acquiring information on service locations and availability [19, 33, 35, 38, 45, 49, 51]. While waiting for services for their child, the immigrant participants in Millau and colleagues' study accessed less external support than the Canadian-born participants, largely because the Canadian caregivers had more contacts and knowledge about available services [41]. Some caregivers also identified the requirement to complete excessive and complex paperwork as a challenge in accessing services [45, 49, 51].

The barriers to accessing *adequate* services included services not providing emotional support [19], a decrease in quality and quantity of services following the end of specialized early intervention [46], and providers' limited autism knowledge [17, 19, 32]. Transition points between services, such as a child moving from early intervention to school, also created difficulties due to poor communication between organizations, limited or poor quality of services, and the child not being prepared [46]. Additionally, some participants disagreed with the types of language or labels used by Canadian professionals, resulting in contention between these groups. For example, some of the caregivers in Pondé and Rousseau's study rejected the notion that their child had any "problem" and instead felt the issue was the lack of acceptance of their children's differences [42]. They also felt that the professionals sending their children for assessments due to their functioning at school was intrusive [42]. This is significant in that the way families conceptualize and approach autism in their daily lives influences their pursuit of therapies [43].

Dissatisfaction with their child's education was also addressed by some caregivers across studies [32, 38, 46, 48]. Poor communication between the school and early intervention services as well as limited staff presented barriers for participants in Rivard and colleagues' study [46]. Some participants felt the staff at their child's school lacked proper training and protocols and received requests to pick their child up from school during the day [32, 38], resulting in some having to leave their jobs [32]. Additionally, some participants in Shanmugarajah and colleagues' study felt that the education system overall was not meeting the needs of their child [48].

**3.3.b. Barriers resulting from immigration.** Barriers related to immigration were the most common themes observed, emerging in 14 of the included studies [17–20, 31–33, 35, 37, 38, 48–51]. They included difficulties due to cultural differences [18, 20, 31, 37, 48, 50], as well as language or communication barriers between caregivers and service providers [17–20, 32, 33, 35, 37, 38, 48–51]. Moreover, although all participants included in Rivard and colleagues' study were fluent in English or French, 14.7% of the participants felt that that other caregivers in their position should prioritize learning their host country's language [45], which suggests that even though their participants did not personally experience language barriers, they recognize how such a barrier could pose problems for newcomer caregivers seeking autism services.

**3.3.c. Personal barriers.** Some of the studies indicated that their participants encountered "personal barriers," meaning internal experiences or events concerned largely with caregiver mental health that may not be apparent to anyone but the caregivers themselves. These were often fueled by external obstacles such as struggling to access services or being in a new environment [37, 40, 43, 49, 51, 52]. One such barrier was the experience of stress [37, 40, 49]. Participants in Pondé and colleagues' study noted feeling pressure from their child's school and healthcare professionals to meet their expectations of everything they must do to support their child, as well as from the demands their child may place on them, such as resistance to completing different tasks [43]. Caregivers also experienced emotional strain and difficulties [17, 30, 33, 37–39, 43, 49, 51], including depression [42, 49, 51] and guilt [17, 37, 43, 49]. Weiss and colleagues found that immigrant mothers of autistic children in Canada had a higher likelihood of feeling unable to meet their child's needs than mothers from other demographic groups [52].

**3.3.d. Social barriers.** Social barriers, such as stigma and discrimination, were identified in 14 studies [17–19, 30–32, 34, 38, 41, 45, 48–51]. Caregivers experienced discrimination based on their minority status (including having an accent) [31, 49, 50], and others felt discriminated against or not accepted due to their child's disability [30, 31, 51]. Moreover, some caregivers came from cultures wherein stigma exists toward people with developmental disabilities [19, 34, 37, 38, 45, 48, 49, 51]. Others were afraid their child would encounter stigma due to being autistic [17, 32, 37, 45], or that they themselves would be stigmatized [32, 45]. Some studies noted autistic children being excluded from school [17, 18].

Many participants also experienced a lack of social support [18, 19, 34, 41, 45, 48, 49, 51]. Reasons for this included immigrating away from their social networks [18, 19, 45, 48, 49, 51], their spouse working outside of the home to support their family [19], immigrating to Canada prior to their spouse [34], and family or community members not understanding or accepting autism [45, 48, 49]. Some caregivers did not disclose their situation to their extended families [51]. Others reported difficulties trusting service providers [18, 37, 49].

**3.3.e. Financial barriers.** Financial barriers were identified by participants in 9 studies [17, 19, 31, 32, 35, 45, 48, 49, 51]. These were perpetuated by a lack of government funding [32, 35, 45, 48, 51], service costs [17, 19, 38, 49, 51], as well as unemployment within the family [38, 49, 52]. One participant in Decoteau's study felt that even though the government funds early intervention services, Somali caregivers do not get access to this funding [31]. Skrinda's study

identified how limited service funding within school boards can also act as a barrier that may prompt families to switch schools [49].

**3.3.f. Social/emotional/personal supports as facilitators.** Social facilitators, conceptualized in this review as social and emotional supports that improved caregiving experiences, coping, or quality of life, were identified in 12 studies [17, 32–34, 36, 38, 41, 45, 48–51]. These included having caregiver(s) of autistic children in their social networks [45, 49], and community [17], as well as networking with other caregivers of autistic children [33, 38, 49–51], friends [32, 38, 51], and family [17, 38, 41, 48, 49, 51].

The mobilization of personal resources was also noted as being beneficial to caregivers. One example of a personal resource that facilitated caregiver wellbeing included faith and religion [32–34, 38, 48]. Actively and/or proactively seeking additional information regarding services and supports, as well as information about autism itself [38], was also identified as an activity caregivers used to cope with their child's diagnosis [32, 33], Additionally, Camard and colleagues explored how immigrant parents took on advocate roles in response to encountering barriers as they navigated their lives with an autistic child [30].

**3.3.g. Service personnel-based facilitators and financial facilitators.** Service providers who were especially helpful acted as facilitators in numerous studies [20, 32, 34, 38, 45–49, 51]. Additional facilitators provided by service personnel included access to information [32, 45, 50], access to services prior to diagnosis [45], satisfaction with the availability of services [48, 51], collaboration and communication with parents and other organizations working with the child [46], and satisfaction with the adequacy of services [20, 38, 44, 48]. Some studies also highlighted the positive impact of services that address immigration-specific barriers. For example, interpreter services were identified as helpful or preferential by Rivard and colleagues and Su and colleagues [45, 51]. Participants in Fong and colleagues' study noted how respect from professionals, which involved taking steps to eliminate service barriers for newcomers, greatly impacted their overall family quality of life [34]. Some participants noted benefits from their service provider sharing their background [35, 37, 48]; one Chinese mother in Lee and Zhu's study felt her understanding of her autistic son's situation was enhanced via the enriching conversations she was able to have with her Chinese-Canadian doctor [37], while a participant in Grewal's study felt that their service provider, also being South Asian, would be more discrete, given their understanding of the stigma around mental health in their community [35]. Government funding was also noted by immigrant caregivers in some studies to facilitate their child's access to some services [32, 44, 45, 51], as well as funding from other institutional sources [38].

**3.3.h. Gender differences.** Lastly, differences between mothers and fathers were identified in five studies [21, 40–42, 45]. For example, Millau and colleagues found that the immigrant fathers were more likely than the Canadian-born fathers to have full-time work and, conversely, less likely to stay home [40]. In terms of how perceptions of autism differed between newcomer mothers and fathers, Pondé and Rousseau found that fathers focused more on the difficulties their child experienced and appeared more unsettled by them than mothers [42]. Millau and colleagues noted differences between mothers and fathers in causal attributions of autism, the proportion that recognized early signs of autism, and the order in which treatments were prioritized [21]. Additionally, when looking at the raw data in the study by Rivard and colleagues, 42.9% of immigrant mothers noted experiencing social isolation compared to 7.7% of included immigrant fathers, and 28.6% of immigrant mothers struggled with denial compared to 7.7% of immigrant fathers [45].

Gender differences were also observed by Millau and colleagues regarding the importance of different subscales of family quality of life, with immigrant fathers ranking disability-related support as being the most important, while immigrant mothers ranked family interaction as

most important [41]. Moreover, when comparing the experiences of immigrant caregivers and Canadian-born caregivers, Millau and colleagues found that while both immigrant mothers and fathers' levels of satisfaction with the family interaction subscale of family quality of life was lower than that of Canadian-born caregivers, the lowest rankings were among immigrant fathers [41].

In summary, studies identified many overlapping barriers and facilitators experienced by newcomer caregivers of autistic children, ranging from those impacting mental health and social wellness to service access and finances. Moreover, differences in the experiences of newcomer mothers and fathers were also identified. Examining these findings further, and the recommendations made within study reports, reveals ways to address gaps in Canadian research on this population.

## 4. Discussion

### 4.1. Key findings

This review examined the aims of the included studies to ascertain what themes researchers have and have not examined pertaining to newcomer caregivers of autistic children in Canada. The three most common aims were to examine caregiver perceptions of and feelings toward autism, to learn more about these populations' experiences during the diagnostic process, and to inquire about the presence of stressors and challenges. Given the well-established importance of early intervention for autism [10, 21, 31, 49], and the heightened risk of culturally and linguistically diverse families not obtaining these services [9], there is a need to understand what is happening around the point of diagnosis and early intervention to ensure that these processes run smoothly. This concept also highlights the importance of examining other factors, such as treatment priorities and intervention usage, which do not appear to be as comprehensively examined in these newcomer studies.

In terms of examining service access, the included studies sought to assess the existence of barriers more often than the existence of facilitators. Many of the barriers identified among the included studies, such as fragmented services, difficult administrative processes, and language barriers, were consistent with barriers that have been identified in pre-existing reviews, thus providing further evidence of their existence for newcomer caregivers in Canada. Although not explicitly stated, immigrant participants in Millau and colleagues' study likely accessed less external support due to barriers in knowing where and how to access local services [41]. In addition to identifying barriers to service access, the studies made a number of overlapping suggestions for ways to address these barriers. Examples of these suggestions include adjusting funding based on familial need [49, 51], improving access to information about autism [19, 31, 45], helping caregivers learn about local social support opportunities [35, 38], increasing professional training on cultural competency/awareness and equitable service provision [35, 37, 44, 47, 49, 51], further provision of culturally sensitive, family-focused interventions [39], funding interpretive support services [19, 21, 49, 51], and ensuring caregivers are aware of what services and funding are available to them [35, 49, 51].

The extent to which caregivers rely on support from organizations depends on factors such as how much support they receive from their families as well as their familiarity with the host culture [54]. Given that newcomer families may leave support networks behind in their home countries [18, 19, 45, 49, 51], as well as how cultural differences can interfere with service access [18, 20, 31, 37, 50], immigrant caregivers may be overly reliant on external services for support. This intensifies the significance and urgency of acknowledging and implementing these suggested recommendations, as well as the need to evaluate such interventions in future research to ensure service access disparities are adequately addressed.

While facilitators were not featured as aims of the included studies to the same extent as barriers, examining what contributes to both social and personal barriers and effective social facilitation could demonstrate the significance of social support and parent mental health to this population [51]. This could be indicative of another area of future research.

Some of the studies also provided insight into the significance of government funding for families in acquiring much-needed services [44, 45, 51]. This finding supports the recommendations made by many of the studies to ensure adequate funding is available for these families [49, 51].

When looking at which studies highlighted barriers and which highlighted facilitators, readers may notice some contradictory overlap. For example, some of the participants in Skrinda's study noted positive interactions with helpful service personnel whereas others described negative interactions [49], hence the inclusion of results from this study in both categories. The same can be said for Lee and Zhu's work [37]. Both of the aforementioned studies had small sample sizes, which points to a greater limitation of conclusions drawn from this review, as discussed below. However, if this same division is observed among studies with larger sample sizes, such as Rivard and colleagues' [45], it may indicate the need for the establishment of more consistent service provision in Canada.

## 4.2. Research gaps

**4.2.a. Lack of refugee studies.**   Originally, the research team intended to examine findings on immigrant and refugee caregiving experiences separately, but only the studies by Decoteau and Skrinda included refugee participants [31, 49]. While only 9,200 refugees were resettled in Canada in 2020, this number was likely influenced by the COVID-19 pandemic as 30,100 refugees were resettled in Canada in 2019, or the year prior to the onset of the pandemic [55]. Limited refugee representation in Canadian autism studies could be due to a lack of prevalence of autism among the refugees entering the country, however, this review highlighted additional factors that can influence study participation, such as recruitment strategies, cultural differences in the understanding of (or education around) autism, or caregivers being uncertain of where to access services. Moreover, studies may not be distinguishing refugees separately from minorities, which has been noted in the past [15]. As such, in order to properly ascertain the level of need among refugee caregivers of autistic children, it is worthwhile to conduct future studies that specifically seek out their participation.

**4.2.b. Lack of studies on the experiences of fathers.**   Another considerable research gap is the lack of research on newcomer fathers acting as caregivers for autistic children. As previously mentioned, none of the included studies focused explicitly on fathers, and in all cases where fathers were included, they were considerably outnumbered by mothers. For example, in Grewal's study, 87.5% of the participants were mothers [35], in Millau and colleagues' it was 62.2% [21], and Rivard and colleagues' it was 61.7% [45]. There are a number of possible reasons for this gap, such as mothers being more likely to leave their jobs to provide full-time care for their developmentally disabled child [19], as well as the way some cultures may influence parental roles; in families following traditional gender roles, fathers may have fewer daily interactions with their children [40]. Even so, the inclusion of fathers and other male caregivers is necessary to both support or refute these reasons, as well as to ensure their support needs are adequately met.

Overall, there is also limited data on the composition of refugee families coming into Canada. This means that, when it comes to this group in particular, a lack of fathers participating in research such as that reviewed in this paper could be due to a lack of fathers within refugee families entering Canada to begin with. Limited information regarding family composition,

and by extension the support available to refugee caregivers of autistic children within their own family unit, points to another direction for future research.

**4.2.c. Methodology.** The majority of the studies reported qualitative data, namely interviews. While these accounts provide insight into the caregiving experiences of participants, these data are not easily generalizable [10]. This highlights the need to generate further quantitative data examining the barriers, facilitators, and general caregiving experiences encountered by newcomers so as to understand their generalized experiences and needs, consequently assisting with the design of broad-based recommendations. This review also establishes that there is some overlap between qualitative findings across studies in terms of barriers and facilitators, providing a potential starting point in the design of future qualitative or mixed-methods studies.

**4.2.d. Representation of Canada's immigrant population.** To examine how representative the current Canadian literature is of the demographics (in this case, country of origin) of the newcomer population, the percent occurrence of specific country of origin from the studies was compared to data on the place of birth of immigrants who arrived in Canada between 2001–2021 in Fig 2 (based off of a dataset published in 2023 by Statistics Canada, see [53]). There are some limitations with the Canadian dataset selected; the participants in the included studies were both immigrants and refugees, whereas the dataset only includes information on immigrants. Moreover, although the included studies were published as of the year 2000, recent immigrants whose data were collected could have arrived in Canada in the preceding years, whereas the Canadian data sample included has date of arrival beginning at the year 2001. Despite these limitations, the authors chose to offer this comparison to provide a sense of whether certain groups were being over or underrepresented in the literature. The chart demonstrated a high degree of Asian participants in the Canadian literature, which approaches the high proportion of Asian immigrants to Canada; the same was true for European and African immigrants. Meanwhile, immigrants from the Caribbean, Latin America, and North America may be overrepresented in the literature compared to Canadian immigrant population proportions. Finally, immigrants born in the Middle East and South America appeared underrepresented compared to the Canadian population proportions, suggesting more research attention to these groups is especially warranted.

## 4.3. Limitations

Many of the included studies had smaller sample sizes and focused on specific ethnic groups. This, in addition to the majority-use of qualitative methods, means that the findings are difficult to generalize. Participation in cross-cultural research can be further limited due to language barriers [11, 54], cultural barriers and geographic location [11], as well as fear, lack of trust, and misunderstanding [54]; these factors can further limit research among these populations.

## 5. Conclusion

This review was conducted to provide an overview of existing Canadian research on newcomer caregivers of autistic children, emphasizing service access and navigation. This review yielded 28 studies, which were primarily qualitative and with data collected in English. While immigrants were represented from all populous continents short of South America, refugee participants were scarcely represented. These studies reaffirmed several of the barriers to services for immigrant caregivers that previous, non-Canadian-specific research has identified, examples being fragmented services and the existence of language barriers. The role that social and emotional support plays in barriers, facilitators, stressors, and buffers was also highlighted. This

review identified several gaps in current Canadian research, namely when it comes to male and refugee caregivers of autistic children. This review also identified the need for more studies that employ quantitative approaches to data generation for these populations.

Sritharan and Koola said it best: "[t]he goals should be to provide culturally sensitive approaches to decrease alienation based on immigrants' personal beliefs, while helping the immigrant community gain better access and early intervention" [10]. This could be extrapolated to the experiences of refugee caregivers of autistic children, as well. Addressing the current gaps in research ensures the needs of newcomer caregivers are being adequately met, which then may positively influence their autistic children's outcomes.

## Supporting information

**S1 Appendix. Search strands.**
(DOCX)

## Acknowledgments

The authors would like to acknowledge the valuable contributions of research assistants Christian Awad with study retrieval and the conduct of the literature searches and AJ Ware with the study screening and theme extraction.

## Author Contributions

**Conceptualization:** Stephen J. Gentles, Janet McLaughlin, Margaret Schneider.

**Data curation:** Lina Marie Casale.

**Formal analysis:** Lina Marie Casale.

**Methodology:** Stephen J. Gentles.

**Supervision:** Stephen J. Gentles, Janet McLaughlin, Margaret Schneider.

**Writing – original draft:** Lina Marie Casale.

**Writing – review & editing:** Lina Marie Casale, Stephen J. Gentles, Janet McLaughlin, Margaret Schneider.

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
