## [Decision Letter · Decision Letter 0]

13 Apr 2023

PONE-D-22-26125Service access experiences of immigrant and refugee caregivers of autistic children in Canada: A scoping reviewPLOS ONE

Dear Dr. Casale,

Thank you for submitting your manuscript to PLOS ONE. After careful consideration, we feel that it has merit but does not fully meet PLOS ONE’s publication criteria as it currently stands. Therefore, we invite you to submit a revised version of the manuscript that addresses the points raised during the review process.

We look forward to receiving your revised manuscript.

Kind regards,

Adetayo Olorunlana, Ph.D.

Academic Editor

PLOS ONE

Journal Requirements:

2. We noticed that the database search of your scoping review was performed in early 2021. Please ensure that your search is up to date and any relevant studies published since early 2021 are included in your review.

Reviewers' comments:

Reviewer's Responses to Questions

**Comments to the Author**

1. Is the manuscript technically sound, and do the data support the conclusions?

Reviewer #1: Yes

Reviewer #2: Yes

2. Has the statistical analysis been performed appropriately and rigorously? 

Reviewer #1: Yes

Reviewer #2: I Don't Know

3. Have the authors made all data underlying the findings in their manuscript fully available?

Reviewer #1: Yes

Reviewer #2: No

4. Is the manuscript presented in an intelligible fashion and written in standard English?

Reviewer #1: Yes

Reviewer #2: Yes

5. Review Comments to the Author

Reviewer #1: This review collects findings from a curated collection of papers investigating the experience of newcomers to Canada pertaining to ASD-specific services, interventions, and related personal experience. The review details selection criteria for the inclusion and exclusion of papers, provides description of features of those included, and distills a set of themes cutting across the papers.

This review’s selection criteria are clear and easy to follow, and overall the manuscript is clear and readable. While the scope is limited to Canada, most of the considerations will be thought-provoking to audiences in other countries. Given the attention to issues of representation in research and clinical service, the demonstrative value of this review has value beyond its specific scope.

I would generally recommend this review for publication, though I include a few minor suggestions for possible improvement:

1. An inherent feature of scientific literature is the influence of prior publications onto others. When identifying general themes, as this review does, the lack of independence of the underlying papers can cause certain themes to rise up due upstream ideas percolating down into later work.

With this in mind, I would recommend the authors consider inspect the papers in some sort of chronological graph indicating linkages and themes (essentially a labeled flow-chart). This need not be a supplemental figure, but could be if it is productively illustrative.

With these connections in hand, I would hope the authors could review their themes and perhaps observe whether some that have appeared in multiple papers did so independently - a mark of general importance above and beyond a mere “sticky” idea from an early paper, or if a less widely-cited theme has only recently cropped up (a possible mark of emerging importance). This seems like a useful means of triaging the themes in the review above and beyond a mere citation count.

2. The review summarizes demographic information from the cited papers (e.g. Figure 2), but could, in theory relate this information to broad immigrant / refugee demographics across Canada from other sources across the highlighted time frame. Given that the emerging consensus is that ASD prevalence (if not diagnosis) is uniform across ethnicities / countries of origin, a comparison of the published studies to broader demographics might highlight gaps in the space of the literature reviewed here.

For a while in the United States there existed a lower rate of ASD in the latinx population, generally thought to be driven by a reduced rate of recognition/diagnosis. Without this first step of identification, these populations would be underrepresented in a review such as this.

A comparison of general population demographics to those of the cited papers might cast light upon those populations currently overlooked in the literature.

3. I was left wondering about the demographic statistics of refugee families. My unfortunate suspicion is that a significant percentage of refugee families entering Canada are incomplete (e.g. missing a father). This seems like a potential for interaction with the larger theme of paternal involvement in caregiving with refugee status. This might be an informative point for discussion toward the particularities of the refugee experience and useful for future investigators.

I don't consider the above suggestions mandatory for acceptance, but if they are fruitful without sacrificing the overall clarity and readability, their inclusion might be of value.

Reviewer #2: Thank you for the privilege of reviewing your scoping review on immigrant and refugee caregivers of autistic children in Canada. I have some concerns about the search reporting. Please follow PRISMA for searching http://www.prisma-statement.org/Extensions/Searching.

1) please include the vendor name for all databases.

2) in the appendix please include the search strategies for all databases, copied and pasted exactly as run.

3) please use the most recent PRISMA flow diagram (2020). In the first box, please provide a breakdown of number of results per database.

4) as there were so few results, I would suggest that in your search, the phenomenon concept was unnecessarily focused and might result in missing relevant citations. I would omit that section of the search and screen for those concepts manually. A search for a scoping review should be broad to capture all available research.

5) For population concepts 1 and 2 you have done a good job of using both subject headings and keywords. However, you did not do this in the other sections of your search. For population concepts 3a and b, please supplement the keywords with the appropriate subject headings. For example, exp Indians, North American/

Inuit/ exp "Emigrants and Immigrants"/ Refugees/ "Transients and Migrants"/ . Similarly for population concept 4, there is a subject heading for Canada: exp Canada/

6) For the dates searched, please provide day and month of searches, as database results vary sometimes daily. Searches should be less than a year old. Your searches are a couple of years old. Please update your searches.

7) For screening of titles/abstracts, how many screeners did you have? There should be two independent reviewers. Similarly for data extraction, this should be done by two independent extractors.

6. PLOS authors have the option to publish the peer review history of their article (what does this mean?). If published, this will include your full peer review and any attached files.

Reviewer #1: No

Reviewer #2: **Yes: **Ani Orchanian-Cheff

---

## [Author Response · Author response to Decision Letter 0]

12 Oct 2023

Please see attached document titled "Response to Reviewers" wherein the comments from each reviewer/editor have been addressed.

---

## [Decision Letter · Decision Letter 1]

18 Oct 2023

Service access experiences of immigrant and refugee caregivers of autistic children in Canada: A scoping review

PONE-D-22-26125R1

Dear Dr. Casale,

We’re pleased to inform you that your manuscript has been judged scientifically suitable for publication and will be formally accepted for publication once it meets all outstanding technical requirements.

Kind regards,

Adetayo Olorunlana, Ph.D.

Academic Editor

PLOS ONE

Additional Editor Comments (optional):

Reviewers' comments:

Reviewer's Responses to Questions

**Comments to the Author**

1. If the authors have adequately addressed your comments raised in a previous round of review and you feel that this manuscript is now acceptable for publication, you may indicate that here to bypass the “Comments to the Author” section, enter your conflict of interest statement in the “Confidential to Editor” section, and submit your "Accept" recommendation.

Reviewer #2: All comments have been addressed

2. Is the manuscript technically sound, and do the data support the conclusions?

Reviewer #2: Yes

3. Has the statistical analysis been performed appropriately and rigorously? 

Reviewer #2: I Don't Know

4. Have the authors made all data underlying the findings in their manuscript fully available?

Reviewer #2: Yes

5. Is the manuscript presented in an intelligible fashion and written in standard English?

Reviewer #2: Yes

6. Review Comments to the Author

Reviewer #2: Thank you for the revisions. I am satisfied with the changes. The searches were rerun according to my comments.

7. PLOS authors have the option to publish the peer review history of their article (what does this mean?). If published, this will include your full peer review and any attached files.

Reviewer #2: **Yes: **Ani Orchanian-Cheff

---

## [Editor Report · Acceptance letter]

31 Oct 2023

PONE-D-22-26125R1 

Service access experiences of immigrant and refugee caregivers of autistic children in Canada: A scoping review 

Dear Dr. Casale:

I'm pleased to inform you that your manuscript has been deemed suitable for publication in PLOS ONE. Congratulations! Your manuscript is now with our production department. 

Kind regards, 

on behalf of

Associate Professor Adetayo Olorunlana 

Academic Editor

PLOS ONE